# Effects of Cooperative-Learning Interventions on Physical Education Students’ Intrinsic Motivation: A Systematic Review and Meta-Analysis

**DOI:** 10.3390/ijerph17124451

**Published:** 2020-06-21

**Authors:** Carlos Fernández-Espínola, Manuel Tomás Abad Robles, Daniel Collado-Mateo, Bartolomé J. Almagro, Estefanía Castillo Viera, Francisco Javier Giménez Fuentes-Guerra

**Affiliations:** 1Faculty of Education, Psychology and Sport Sciences, University of Huelva, 21071 Huelva, Spain; carlos.fernandez@ddi.uhu.es (C.F.-E.); bartolome.almagro@dempc.uhu.es (B.J.A.); estefania.castillo@dempc.uhu.es (E.C.V.); jfuentes@uhu.es (F.J.G.F.-G.); 2Centre for Sport Studies, Rey Juan Carlos University, 28943 Madrid, Spain; danicolladom@gmail.com

**Keywords:** cooperative learning, self-determination theory, achievement goal theory, school-based interventions, teacher education

## Abstract

The aim was to review the effects of cooperative learning interventions on intrinsic motivation in physical education students, as well as to conduct a meta-analysis to determinate the overall effect size of these interventions. The PRISMA guidelines were followed to conduct this systematic review and meta-analysis. The PEDro Scale was used to assess the risk of bias and the GRADE approach was used to evaluate the quality of the evidence. A total of five studies fulfilled the inclusion criteria and they were included in the meta-analysis. Effect size for intrinsic motivation of each study was calculated using the means and standard deviations of the Perceived Locus of Causality Scale (PLOC) before and after the intervention. The overall effect size for intrinsic motivation was 0.38 (95% CI from 0.17 to 0.60) while the heterogeneity was large. Although four of the five studies reported significant within-group improvements in intrinsic motivation, only three studies showed significant between-group differences in favor of the experimental group. The findings showed that program duration and participant age may be relevant factors that must be considered by educators and researchers to conduct future effective interventions. Cooperative learning interventions could be a useful teaching strategy to improve physical education students’ intrinsic motivation. However, given the large heterogeneity and the low quality of the evidence, these findings must be taken with caution.

## 1. Introduction

Over the last two decades, achievement goal theory [1] together with the self-determination theory [2,3] and Vallerand’s hierarchical model of motivation [4] have been effectively complemented to examine physical education students’ motivational process. These last two theories describe a sequence of motivational variables where social factors (i.e., the teacher’s interpersonal style) can influence satisfaction or can thwart basic psychological needs (autonomy, competence and relatedness). Autonomy can be defined as a person’s need to experience actions of their own choosing. Competence refers to the need to control results and achieve efficiency; relatedness can be defined as the need to be connected to others and the feeling of being accepted [3]. In this regard, the satisfaction of these needs has been linked to some types of self-determination motivation, such as intrinsic motivation. Intrinsic motivation is defined as “spontaneous activity that is sustained by the satisfactions inherent in the activity itself” [3] (p. 99). Therefore, intrinsic motivation involves people participating freely in activities that they find interesting, that provide novelty, fun and an appropriate challenge [2]. In consequence, it can help to achieve benefits in the cognitive, affective and psychomotor aspects of physical education students [5,6].

Resembling the teacher’s interpersonal style, the motivational climate generated by a physical education teacher has been studied in the scientific literature as one of the social factors in the motivational sequence described above (social factors → basic psychological needs → types of motivation → consequences). The motivational climate is a construct from the achievement goal theory [1]. According to this theory, Ames [7] postulated that social agents (teacher, peers and parents) build two possible motivational climates: a task-oriented climate (where personal growth, effort and skill progression are highlighted); and an ego-oriented climate (where comparison between peers, member rivalry and results are emphasized). The scientific literature has shown that a task-oriented climate is linked to satisfaction of the three needs [8,9] and consequently it is also related to the most self-determined types of motivation [10].

In relation to achievement goal theory, cooperative learning has been suggested as a teaching strategy that can improve the motivation of young people [11]. A possible relation among a task-oriented climate and cooperative structure was hypothesized because they both highlight working and effort with others, instead of against others [12]. In fact, one of the most important instruments used in the scientific literature to measure these two motivational climates in sport and physical education contexts, the Perceived Motivational Climate in Sport Questionnaire-2 by Newton, Duda and Yin [13], considers that cooperative learning along with effort and role constitute the three subscales that make up the task-oriented climate dimension. This questionnaire has been used in previous studies conducted in different countries and focused on the evaluation of motivation in physical education students [14,15,16].

Cooperative learning is considered a pedagogical model, which can help to achieve learning outcomes of four types: physical, affective, social and cognitive [17]. For many years, cooperative learning has been used in the context of physical education [18,19]. In these studies, cooperative learning interventions were conducted in different educational settings, reporting positive effects on motor skills, social skills, cognitive understanding and on creating an affective domain for physical education students. In this sense, this pedagogical model could be a useful teaching strategy to improve intrinsic motivation since it could be associated with the satisfaction of basic psychological needs from self-determination theory. This model can help to satisfy the need of relatedness since it can develop good social relations between peers [20] and can satisfy the need of competence by increasing physical skills [21]. Precisely, this fact has been analyzed in other school subjects. In the study by Hänze and Berger [22], cooperative learning improved the expertise of basic needs in physics classes, which determined more intrinsic motivation in students. Accordingly, it is possible to hypothesize that a similar process may happen in the subject of physical education. In fact, a recent systematic review of cooperative learning in physical education [23] has showed that of the four types of outcomes, social learning has been the most frequently assessed in the last five years, focusing on the relationships between teachers and students and motivation.

Therefore, the main aim of this article was to conduct a systematic review of the scientific literature on the effects of cooperative learning interventions on physical education students’ intrinsic motivation. For this purpose, the effect sizes of cooperative learning interventions on intrinsic motivation were determined through a meta-analysis.

## 2. Materials and Methods

The Preferred Reporting Items for Systematic Review and Meta-analysis Protocols (PRISMA) guidelines have been followed to carry out the current systematic review [24].

### 2.1. Literature Search

The Web of Science (WOS), Scopus and SportDiscus electronic databases were utilized to locate the studies selected in the present systematic review with meta-analysis. The search terms were divided into four groups of keywords as follows: (1) “cooperative learning”, “cooperative games”, “models-based learning”; (2) “intervention”, “experimental”, “quasi-experimental”, “implementation”, “randomized controlled trial”; (3) “motivation”; and (4) “physical education”. The Boolean operator “or” was incorporated between the words included in the first and second groups and the operator “and” was incorporated between each group. The manuscript search was conducted in April 2020.

### 2.2. Study Selection

The manuscripts were selected if they fulfilled the following inclusion criteria: (a) intervention was based on cooperative learning, (b) physical education students’ intrinsic motivation was measured, (c) the article was written in English or Spanish, (d) the manuscript was an article published or accepted in a peer review journal. No inclusion criteria related to the years of publication of the articles were considered. The study selection process was conducted by two independent authors (C.F.-E., and M.T.A.R.).

### 2.3. Assessment of Risk of Bias

To assess risk of bias, the PEDro scale was used [25]. This scale was developed to evaluate the quality of intervention studies, especially randomized controlled trials. The GRADE guidelines, which involves a four-point scale (“very low”, “low”, “moderate” and “high”) was used to assess the quality of evidence [26]. Table 1 shows the risk of bias results of included articles.

### 2.4. Data Collection

Firstly, two authors extracted data from the included articles. Subsequently, another author checked the extracted information. In accordance with the recommendations from PRISMA guidelines, the information extracted was as follows: participants, intervention, comparisons, outcomes or results and study design (PICOS) [27]. Table 2 shows the main characteristics of the different protocols of intervention and the main participants’ characteristics: sex, age, level of education and sample size. Regarding interventions, Table 3 summarizes the following details: duration of the study, number of sessions and type of cooperative intervention program. 

### 2.5. Statistical Analysis

For this meta-analysis, a random-effects model was used to measure the effect of cooperative learning interventions on intrinsic motivation in physical education students. The results of each study on this variable can be seen in Table 3. The treatment effect was calculated as the difference between the change in the experimental group and the change in the control group. In each study, effect size was calculated using the means and standard deviations before and after treatment [28]. The magnitude of the Cohen′s d was identified as follows: (a) “large”, for values higher than 0.8, (b) “moderate”, when it was between 0.5 and 0.8, (c) “small”, for values between 0.2 and 0.5, and (d) “no effect” for values below 0.2. Heterogeneity was assessed by calculating the following statistics: (a) Tau^2^, for the calculation of variance between studies, (b) Chi^2^ and (c) I^2^, which is a transformation of the H statistic used to determine the percentage of the variation that is caused by the heterogeneity. The most common classification of I^2^ considers values higher than 50% as large heterogeneity, values between 25% and 50% as average and lower than 25% as small [29]. The tool Review Manager 5.3 was used to conduct all analyses [30].

## 3. Results

### 3.1. Study Selection

The complete process (PRISMA flow diagram) of this review is shown in Figure 1. A total of 27 records were identified in the electronic databases—WOS (13), Scopus (7) and SportDiscus (7)—12 of which were removed because they were duplicated. Of the remaining 15 records, three were eliminated because they were not an intervention, three because the intervention programs were not based on cooperative learning, three because they did not measure the intrinsic motivation and one because it did not use a control group. Finally, five studies were included in this systematic review and meta-analysis after an exhaustive selection.

### 3.2. Risk of Bias

Table 1 shows the risk of bias of the five selected articles according to the PEDro scale. Scores varied from four [31] to seven [32,33]. Three studies showed a lower risk of bias, as they indicated a score of ≥6 [32,33,34]. By contrast, two studies showed a higher risk of bias, as they indicated a score of <6 [31,35]. Regarding the quality of evidence, the GRADE guidelines have been followed. In this sense, the rating start was “low” because the outcomes do not come from randomized controlled trials. Additionally, the quality of evidence was downgraded for reasons of inconsistency due to the degree of heterogeneity. Thus, the quality of evidence according to the GRADE guidelines was “very low”, which was defined as “We have very little confidence in the effect estimate: The true effect is likely to be substantially different from the estimate of effect” [26] (p. 404).

### 3.3. Study Characteristics

Table 2 and Table 3 show a summary of the study characteristics. The total sample was 1020. Of these, 518 were distributed in the cooperative learning group (CLG) and 502 belonged to the control group (CG). Two studies were conducted in Primary School, two in High School and another at University level.

### 3.4. Interventions

Table 2 shows a summary of the cooperative learning interventions in each article: three studies used different cooperative learning structures or techniques such as Coop-Coop, Learning Together, Think-Share-Perform, Team Learning, Collective Score, etc. Only one study did not report information in this sense [35]. This table also summarizes the interventions of the control group of each article: all studies used traditional teaching methods.

Table 3 summarizes the duration of the programs and the number of sessions in each study. Intervention durations varied between three weeks and six months. The number of sessions ranged between 6 and 30. The study conducted by Cecchini et al. [33] did not report the number of sessions. 

### 3.5. Outcome Measures

The effects of cooperative learning interventions on physical education students’ intrinsic motivation are shown in Figure 2. To evaluate motivation levels, all five articles used the Spanish version of Perceived Locus of Causality Scale (PLOC), which is a version of a scale originally written in English by Goudas, Biddle and Fox [36]. This Scale was translated and validated for the Spanish context by Moreno-Murcia, González-Cutre and Chillón [37] and it measures five motivation levels (four items for level): Intrinsic motivation (e.g., “because Physical Education is fun”), Identified Regulation (e.g., “because I can learn new skills that I could use in other areas of my life”), Introjected Regulation (e.g., “I would feel bad about myself if I didn’t”), External Regulation (e.g., “because that is what I am supposed to do”), and Amotivation (e.g., “I really feel I’m wasting my time in Physical Education”). The previous sentence of this scale is “I take part in this Physical Education class”. For the answers, a Likert Scale from 1 (strongly disagree) to 7 (strongly agree) is used.

Having used PLOC to measure the physical education students’ intrinsic motivation, four of the five studies reported a significant increase in this variable relative to the baseline caused by cooperative learning intervention. Conversely, the study by Fernández-Argüelles and González-González de Mesa [31] reported a reduction in intrinsic motivation after the cooperative learning intervention. Nevertheless, the *p*-value was not reported. Regarding control groups, three studies reported non-significant changes and another study did not report them [31].

Interestingly, although three studies reported statistically significant improvement in the experimental group relative to baseline, as can be seen in Figure 2, the studies by Cecchini et al. [33], by Fernández-Río, Cecchini, and Méndez-Giménez [34] and by Fernández-Río et al. [32] showed significant between-group differences in favor of the CLG.

Overall effect size for intrinsic motivation was 0.38 with a 95% CI from 0.17 to 0.60. In accordance with the proposed classification, this effect size was small. The heterogeneity level was high: Tau^2^ = 0.05; Chi^2^ = 19.72, df = 4 (*p* = 0.0006); I^2^ = 80%; test for overall effect: Z = 3.55 (*p* = 0.0004).

## 4. Discussion

This systematic review with meta-analysis aimed to analyze the effects of cooperative learning interventions on physical education students’ intrinsic motivation. The main finding according to meta-analysis results was that cooperative learning could improve the intrinsic motivation in physical education students. Figure 2 shows that the effect size in three of five studies was in favor of the experimental group. These findings are in accordance with self-determination theory [2,3] and Vallerand’s hierarchical model of motivation [4], which described how social factors, where cooperative learning is included, can influence the different forms of motivation, and that this effect is exerted by means of the satisfaction of the basic psychological needs (competence, autonomy and relatedness). This improvement can be considered as small according to the overall effect size (d = 0.38 with a 95% CI from 0.17 to 0.60; *p* = 0.0004). Due to the low quality of the evidence, the interpretation of this meta-analysis must be done with caution. However, findings from the current meta-analysis could be useful for teachers and educators to improve intrinsic motivation in physical education students.

From the five interventions based on cooperative learning included in this meta-analysis, four studies reported a significant improvement in intrinsic motivation. The duration of these effective interventions varied between three weeks and six months [32,33,34,35]. Conversely, in the study by Fernández-Argüelles and González-González [31] a decrease in intrinsic motivation was observed. Given that the intervention’s duration of that study was six weeks, the program duration does not seem to be the most critical factor. However, Figure 2 shows that the three longest studies [32,33,34] reported a higher effect size to the other two studies [31,35]. Therefore, according to this result, cooperative learning interventions should be carried out at least for 12 weeks to achieve a significant and relevant improvement in intrinsic motivation.

An important factor could be the age of the students. The study which reported a reduction in intrinsic motivation had a sample composed of primary school students with a mean age of 8.4 [31]. As shown in Figure 2, this article reported a difference in favor of the control group. The remaining four studies (which reported significant improvement in intrinsic motivation) had a sample composed of primary school students with a mean age of 10.29 [35], high school students with a mean age of 13.66 [32] and 14.60 [33] and university students with a mean age of 20.39 [34]. These results are in line with the findings showed in the study by Hortigüela-Alcalá et al. [38]; they contrasted the effects of a cooperative learning intervention (without a control group) on factors such as motivation in students in two different educational stages: Primary Education (with a mean age of 11.37) and Secondary Education (with a mean age of 15.42). Motivation increased significantly in both groups. In this meta-analysis, the best results were achieved in the studies by Cecchini et al. [33] and by Fernández-Río et al. [32,34] Therefore, although the highest effect size was observed in the study by Cecchini et al. [33], in general, the results show that the effect size was higher in accordance with an increase of the age of participants. Consequently, the lowest effect size was observed in the study by Fernández-Argüelles and González-González de Mesa [31]. Furthermore, Fernández-Argüelles and González-González de Mesa [31] reported that an important limitation of their study was the young age of students, which may impair the execution of the program given the complexity of the strategies carried out in the sessions. Cooperative learning requires a high level of autonomy; thus, it would be advisable and interesting for future cooperative learning interventions with school students to conduct a previous program to develop students’ autonomy before the intervention. For this purpose, a second experimental group could be added in future studies with Primary School students.

Regarding the type of cooperative learning techniques or structures (as seen in Table 2), Fernández-Río et al. [34] used two: Learning Together [39] and Coop-Coop [40]. More cooperative learning structures can be observed in the study by Fernández-Río et al. [32], since they used several techniques as: Think-Share-Perform [41], Co-op Play [42], Collective Score [43], Learning Teams [41]. Learning Groups [44] and Pairs-Check-Perform [41]. Collective Score [43] was also used in the study by Fernández-Argüelles and González-González de Mesa [30]. Although it seems that this study used more cooperative learning structures during the intervention, the authors did not report more details. Finally, neither cooperative learning structure was found in the study by Navarro-Patón et al. [35] nor in the study by Cecchini et al. [33]. Based on the results of this systematic review and meta-analysis and the differences in terms of participants’ age and the program’s duration, it is not possible to determine whether some program types were more appropriate than others. However, an important implication of this study is possibly the inclusion of cooperative learning-based interventions in physical education teacher training programs. In this sense, further studies are needed to know which cooperative learning structures are more effective to improve students’ intrinsic motivation in physical education. In the future, the implementation of high-quality interventions as randomized controlled trials are recommended to obtained results of greater value.

The present systematic review with meta-analysis has various limitations. Firstly, the literature search was limited to two languages: Spanish and English. Therefore, the risk of exclusion of articles written in other languages was high. Secondly, the meta-analysis was conducted with a relatively small sample size, since only five studies fulfilled the eligibility criteria. Thirdly, the meta-analysis shows great heterogeneity and some of the studies analyzed are not of sufficient quality. Therefore, the interpretation of the results from this study must be taken with caution.

## 5. Conclusions

Cooperative learning interventions could be a useful teaching strategy to improve the physical education student’s intrinsic motivation since the overall effect size was significantly in favor of the experimental group. Program duration and participant’s age are relevant factors that must be considered by educators and researchers to conduct future effective interventions. In this sense, a student’s age may impair their execution of the program, given the complexity of the strategies carried out in the sessions. In addition, in terms of the duration of cooperative learning interventions, it is possible for them to achieve a significant and relevant improvement on intrinsic motivation when they last at least 12 weeks. Nevertheless, it is not possible to determine whether some program types are more appropriate than others. Given the very low quality of evidence and the large heterogeneity, these findings must be taken with caution.

## Figures and Tables

**Figure 1 ijerph-17-04451-f001:**
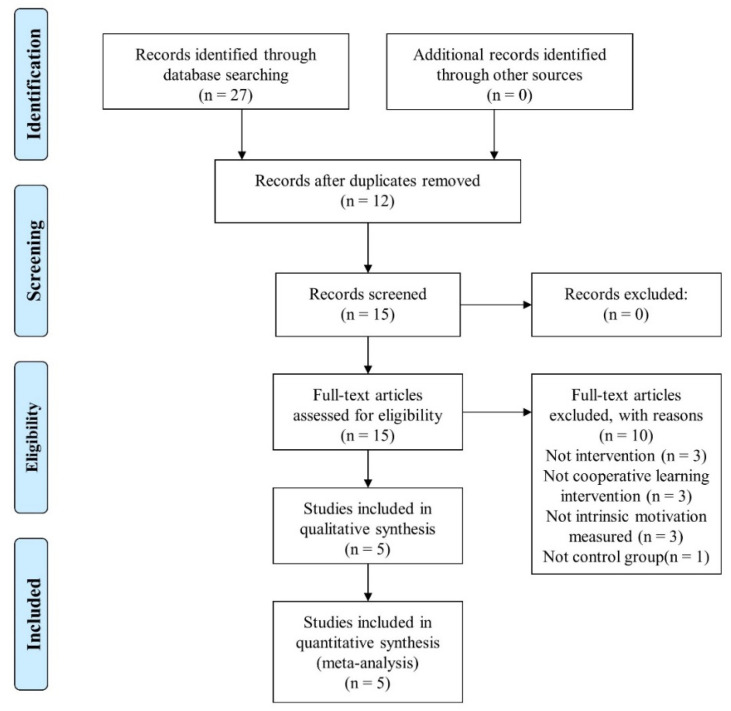
Flow diagram for the systematic review process according with PRISMA statements.

**Figure 2 ijerph-17-04451-f002:**
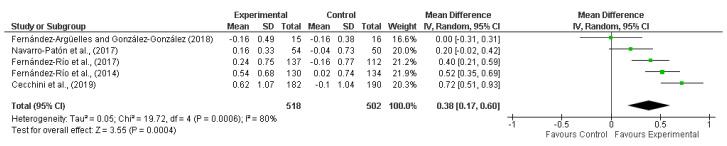
Meta-analysis: effect sizes for intrinsic motivation of cooperative learning interventions.

**Table 1 ijerph-17-04451-t001:** Risk of bias according to the PEDro Scale.

Study	Response to Each Item Level of Evidence											
	1	2	3	4	5	6	7	8	9	10	11	Total Score
Cecchini et al., 2019	Y	N	Y	Y	Y	N	N	Y	Y	Y	Y	7
Fernández-Argüelles and González-González de Mesa, 2018	N	N	N	N	Y	N	N	Y	Y	Y	N	4
Navarro-Patón et al., 2017	Y	N	N	Y	Y	N	N	Y	Y	Y	N	5
Fernández-Río et al., 2017	Y	N	Y	Y	Y	N	N	Y	Y	Y	Y	7
Fernández-Río et al., 2014	Y	N	Y	N	Y	N	N	Y	Y	Y	Y	6

Y: criterion fulfilled; N: criterion not fulfilled; 1: eligibility criteria were defined; 2: the participants were randomly distributed to groups; 3: the assigned was concealed; 4: the groups were similar before the intervention (at baseline); 5: all participants were blinded; 6: therapists (teachers) who conducted the intervention were blinded; 7: there was blinding of all evaluators; 8: the measures of at least one of the fundamental outcomes were attained from more than 85% of the participants initially; 9:” intention to treat” analysis was conducted on all participants who received the control condition or treatment as assigned; 10: the findings of statistical comparisons between groups were reported for at least one fundamental outcome; 11: the study gives variability and punctual measures for at least one fundamental outcome; total score: each satisfied item (except the first) adds 1 point to the total score.

**Table 2 ijerph-17-04451-t002:** Characteristics of the participants and the protocol.

Study	Characteristics of the Sample		Protocol	
	Sample Size of Groups and Sex	Age (SD) and Education Level	Cooperative Learning Group Treatment	Control Group Treatment
Cecchini et al., 2019	CLG: 182 (86 females)	14.20 (2.34) High school	Cooperative learning strategies	Traditional teaching
	CG: 190 (89 females)			
Fernández-Argüelles and González-González de Mesa, 2018	CLG: 16 (6 females)	8.4 (NR) Primary school	Cooperative games with cooperative learning structures	Traditional teaching based on the competition
	CG: 15 (9 females)			
Navarro-Patón et al., 2017	CLG: 54 (24 females)	10.29 (0.62) Primary school	Cooperative games	Traditional teaching
	CG: 50 (21 females)			
Fernández-Río et al., 2017	CLG: 137 (71 females)	13.66 (1.51) High school	Cooperative learning strategies with cooperative learning structures	Traditional teaching
	CG: 112 (56 females)			
Fernández-Río et al., 2014	CLG: 130 (87 females)	20.39 (NR) University	Cooperative learning strategies with cooperative learning structures	Traditional teaching
	CG: 134 (89 females)			

Note: CLG = Cooperative Learning Group; CG = Control Group; SD = Standard Deviation; NR = Not Reported.

**Table 3 ijerph-17-04451-t003:** Characteristics of the interventions (duration and activities).

Study	Duration of Study	Number of Sessions	Type of Cooperative Intervention Program
Cecchini et al., 2019	6 months	NR	Four cooperative learning approaches (conceptual, curricular, structural and complex instruction) were used in the sessions. The program included cooperative learning strategies in sessions of physical expression based on football, volleyball and basketball.
Fernández-Argüelles and González-González de Mesa, 2018	6 weeks	12 sessions	Cooperative games and exercises without competition. Learning structures were based on models-based learning as collective scores were used: the class or different groups should conduct a task whereby several points can be achieved during a determined time. Each small group member obtains an individual score, which is added to a total group score.
		1 h per session	
Navarro-Patón et al., 2017	3 weeks	6 sessions	Within all the cooperative learning units, each session was structured in the following form: an information phase, an activation phase, a goal achievement phase, a cool down phase and a final reflection.
		1 h per session	
Fernández-Río et al., 2017	16 weeks	30 sessions	Within all the Cooperative Learning units, small, heterogeneous working groups were created to capitalize on learning. All units included the five key elements: face-to-face promotive interaction, positive interdependence, individual accountability, interpersonal and small-group skills and group processing. Additionally, five cooperative learning structures were used: (1) Think-Share-Perform: students think, share ideas and negotiate to solve a challenge; (2) Co-op Play: Students should cooperate to solve a challenge; (3) Collective Score: as was explained previously; (4) Learning Teams or learning groups: in groups of four, students divided into different roles (teacher, performer, observer equipment manager) to solve a task or learn a skill; (5) Pairs-Check-Perform: in pairs, students learned a skill divided into two roles: apprentice and teacher.
		1 h per session	
Fernández-Río et al., 2014	12 weeks	24 sessions	Eight cooperative learning strategies were used: (1) classroom’s physical layout was reorganized; (2) heterogeneous and small working groups were created, (3) a minimum time (15 min) was given to each session for working in the small groups; (4) a positive classroom climate to encourage students to participate without fear was created; (5) students could use educational material to study the subject contents; (6) a wide variety of teaching-learning activities of a theoretical and practical nature were utilized; (7) different cooperative learning techniques were used, such as Learning Together (promotion of group activities (e.g., debates) and positives attitudes (e.g., respect, effort or help) or Coop-Coop (the unit is divided in subunits, which are distributed into different groups. Afterwards, each small group member investigates a part of the subunit assigned to finally explain it to her/his classmates); and (8) lastly, assessment tools and different procedures of assessment were used.
		1 h per session	

Note: NR = Not Reported.

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
