# Peer review of "Effects of Cooperative-Learning Interventions on Physical Education Students’ Intrinsic Motivation: A Systematic Review and Meta-Analysis"

_ijerph, 2020, doi:10.3390/ijerph17124451_

Round 1

Reviewer 1 Report

This is a fine contribution to the field. 

The abstract is good. 

Page 1, line 34. Can you be more specific? You note the last decades. Do you mean the last two decades or several? Just add a word to clarify the point. 

line 62, delete nowadays. That word is more appropriate for colloquial speech and not for a scholarly article. 

line 80, PRISM, can you spell out the word and use a capital letter for each, then use the abbreviation. 

Page 4–great figure. 

Pages 4-6 great tables! Very clear presentation of your information. 

Page 8, conclusion is too brief. You need to add 3 more sentences to flesh it out. 

Author Response

COVER LETTER

Manuscript ID: ijerph-823207. Type of manuscript: Article. Title: Effects of cooperative-learning interventions on physical education students´ intrinsic motivation: A systematic review and meta-analysis

Reviewer 1’s comments and suggestions for authors

Details of the revisions and responses

We have added a word in order to clarify this point

line 62, delete nowadays. That word is more appropriate for colloquial speech and not for a scholarly article

We have deleted nowadays

line 80, PRISM, can you spell out the word and use a capital letter for each, then use the abbreviation

We have spelled out the word PRISMA and used a capital letter for each

Page 8, conclusion is too brief. You need to add 3 more sentences to flesh it out

We have expanded the conclusion section

THANK YOU for your comments and suggestions

Reviewer 2 Report

The aim of this study was to review the effects of cooperative learning interventions on intrinsic motivation in physical education students, and to conduct a meta-analysis to determinate the overall effect size of this interventions. Albeit the small number of studies that fit the inclusion criteria, the authors did a great job detailing the study methods and results. The following are a few comments and suggestions for the authors to consider:

Abstract, line 18: The authors wrote, “A total of four studies fulfilled the inclusion criteria and they were included in the meta-analysis.” Aren’t there 5 studies?

Section 2.2 Study selection: Describe the inclusion criteria for the years in which the articles are published

Line 178: Change “Table 3” to “Figure 2” in this sentence: “The effects of cooperative learning interventions on physical education students´ intrinsic motivation are shown in Table 3.”

Line 244: Remove “to” in this sentence, “Regarding to the type of cooperative learning techniques or structures…” Please check throughout the paper for similar corrections.

Discussion: Perhaps provide the implications of this study in terms of physical education teacher preparation program in training preservice teachers and/or in-service teachers on cooperative learning interventions.

Author Response

COVER LETTER

Manuscript ID: ijerph-823207. Type of manuscript: Article. Title: Effects of cooperative-learning interventions on physical education students´ intrinsic motivation: A systematic review and meta-analysis

Reviewer 2’s comments and suggestions for authors

Details of the revisions and responses

We have changed 4 studies for 5 studies (abstract, line 18)

Section 2.2 Study selection: Describe the inclusion criteria for the years in which the articles are published

We have added the following sentence to this: “No inclusion criteria related to the years of publication of the articles were considered”

Line 178: Change “Table 3” to “Figure 2” in this sentence: “The effects of cooperative learning interventions on physical education students´ intrinsic motivation are shown in Table 3.”

We have changed “Table 3” to “Figure 2” in that sentence

Line 244: Remove “to” in this sentence, “Regarding to the type of cooperative learning techniques or structures…” Please check throughout the paper for similar corrections.

We have removed this word throughout the paper for similar expressions

Discussion: Perhaps provide the implications of this study in terms of physical education teacher preparation program in training preservice teachers and/or in-service teachers on cooperative learning interventions.

We have considered this suggestion in discussion section

THANK YOU for your comments and suggestions

Reviewer 3 Report

This is an interesting article on the use of cooperative learning strategies in Physical Education and students' intrinsic motivation.

I suggest that rather than a systematic review with meta analysis, this is changed to a narrative review/systematic review. The meta analysis is based on 5 studies of low quality, leading to a result that is not reliable based on GRADE, and therefore does not add value to the paper. Rather, an in-depth review of the studies, with a recommendation for further high-quality research to be conducted in future, would be of greater interest and value. An expanded version of the current discussion, greater detail of the underlying concepts in the introduction, and description/comparison of the various learning strategies would be welcome.

In the current manuscript, attention needs to be paid to consistency in reporting of numbers (number of studies in abstract, number of studies excluded in the text, number of sessions in one study described as 'not reported' but added into a table).

PEDro scale score meanings are not described in the text. Values below 6 reflect low robustness to bias, which is the case for two of the five studies.

Values for effect size Cohen's d are described as 'small' between 0 and 0.5. Cohen's d is considered to reflect 'no effect' below 0.2.

Page 2, line 68: 'physiological' should read 'psychological'.

Page 7, line 192-194: p value can easily be calculated from the information provided in figure 2 and all are non-significant.

English needs attention throughout to ensure sentences and paragraphs flow more naturally.

Author Response

COVER LETTER

Manuscript ID: ijerph-823207. Type of manuscript: Article. Title: Effects of cooperative-learning interventions on physical education students´ intrinsic motivation: A systematic review and meta-analysis

Reviewer 3’s comments and suggestions for authors

Details of the revisions and responses

The authors thank the suggestion. We are aware of this limitation, but it is due to the design of the studies. The GRADE guideline is designed to clinical studies. In this sense, we have added a suggestion for future studies. We have recommended high-quality research (as randomized controlled trial) should be conducted in future. Furthermore, in the discussion section, we have pointed out this limitation of the study: some of the studies analysed are not of sufficient quality. However, we believe that the meta-analysis carried out can provide important considerations regarding the use of cooperative learning and the promotion of students' intrinsic motivation, although, the interpretation of the results must be taken with caution.

An expanded version of the current discussion, greater detail of the underlying concepts in the introduction, and description/comparison of the various learning strategies would be welcome.

More concepts in the introduction were included.

In the current manuscript, attention needs to be paid to consistency in reporting of numbers (number of studies in abstract, number of studies excluded in the text, number of sessions in one study described as 'not reported' but added into a table).

Thank you for your suggestion. There were errors due to the previous version of the manuscript. Numbers were reviewed and modified.

PEDro scale score meanings are not described in the text. Values below 6 reflect low robustness to bias, which is the case for two of the five studies.

Values for effect size Cohen's d are described as 'small' between 0 and 0.5. Cohen's d is considered to reflect 'no effect' below 0.2.

We have modified the description for effect size Cohen´s d.

PEDro scale score meanings were added in the text.

Page 2, line 68: 'physiological' should read 'psychological'.

We have changed the word 'physiological' to 'psychological'

Page 7, line 192-194: p value can easily be calculated from the information provided in figure 2 and all are non-significant.

Thank you for the suggestion. Only three studies are significant.

English needs attention throughout to ensure sentences and paragraphs flow more naturally.

Sentences and paragraphs were reviewed.

THANK YOU for your comments and suggestions

Round 2

Reviewer 3 Report

This is a much improved version of the manuscript. I still believe you cannot come to a firm conclusion that a cooperative learning programme needs to be at least 12 weeks long, due to the low quality of your data. More cautious language is needed in the sentence itself (something like 'it may be that...' or 'it is possible that...').

The manuscript would benefit from professional proof reading to ensure correct grammar and flow of sentences.

Author Response

COVER LETTER

Manuscript ID: ijerph-823207. Type of manuscript: Article. Title: Effects of cooperative-learning interventions on physical education students´ intrinsic motivation: A systematic review and meta-analysis

Reviewer 3’s comments and suggestions for authors (Round 2)

Details of the revisions and responses

We've used cautious language in the conclusions. “In addition, in terms of the duration of cooperative learning interventions, it is possible for them to achieve a significant and relevant improvement on intrinsic motivation, when they last at least 12 weeks”.

The manuscript would benefit from professional proof reading to ensure correct grammar and flow of sentences.

Sentences have been reviewed in order to ensure correct grammar and flow of sentences.

THANK YOU for your comments and suggestions
